# Assessing Feasibility of Cognitive Impairment Testing Using Social Robotic Technology Augmented with Affective Computing and Emotional State Detection Systems

**DOI:** 10.3390/biomimetics8060475

**Published:** 2023-10-06

**Authors:** Sergio Russo, Letizia Lorusso, Grazia D’Onofrio, Filomena Ciccone, Michele Tritto, Sergio Nocco, Daniela Cardone, David Perpetuini, Marco Lombardo, Daniele Lombardo, Daniele Sancarlo, Antonio Greco, Arcangelo Merla, Francesco Giuliani

**Affiliations:** 1Research & Innovation Unit, Foundation IRCCS Casa Sollievo della Sofferenza, 71013 San Giovanni Rotondo, Italy; s.russo@operapadrepio.it; 2Interdisciplinary Department of Medicine, School of Medical Statistics and Biometry, University of Bari Aldo Moro, 70124 Bari, Italy; 3Clinical Psychology Service, Health Department, IRCCS Casa Sollievo della Sofferenza, 71013 San Giovanni Rotondo, Italy; g.donofrio@operapadrepio.it (G.D.); f.ciccone@operapadrepio.it (F.C.); 4Next2U Srl, Via dei Peligni 137, 65127 Pescara, Italy; m.tritto@next2u-solutions.com (M.T.); s.nocco@next2u-solutions.com (S.N.); 5Department of Engineering and Geology, University G. D’Annunzio of Chieti-Pescara, 65127 Pescara, Italy; d.cardone@unich.it (D.C.); david.perpetuini@unich.it (D.P.); arcangelo.merla@unich.it (A.M.); 6Behaviour Labs S.r.l.s. Piazza Gen. di Brigata Luigi Sapienza 22, 95030 Sant’Agata Li Battiati, Italydaniele.lombardo@blabs.eu (D.L.); 7Geriatrics Unit, Foundation IRCCS Casa Sollievo della Sofferenza, 71013 San Giovanni Rotondo, Italy; d.sancarlo@operapadrepio.it (D.S.); a.greco@operapadrepio.it (A.G.)

**Keywords:** MMSE, social robotics, Pepper robot, human–robot interaction, older adult care, emotional state recognition, cognitive impairment

## Abstract

Social robots represent a valid opportunity to manage the diagnosis, treatment, care, and support of older people with dementia. The aim of this study is to validate the Mini-Mental State Examination (MMSE) test administered by the Pepper robot equipped with systems to detect psychophysical and emotional states in older patients. Our main result is that the Pepper robot is capable of administering the MMSE and that cognitive status is not a determinant in the effective use of a social robot. People with mild cognitive impairment appreciate the robot, as it interacts with them. Acceptability does not relate strictly to the user experience, but the willingness to interact with the robot is an important variable for engagement. We demonstrate the feasibility of a novel approach that, in the future, could lead to more natural human–machine interaction when delivering cognitive tests with the aid of a social robot and a Computational Psychophysiology Module (CPM).

## 1. Introduction

The COVID-19 pandemic highlighted an important need for digital tools. During this period, hospitals, and health systems in general, implemented different strategies to handle the crisis [1]. Especially since the end of the COVID-19 outbreak, the health system has been experiencing a crisis in terms of available human resources, which was foreseen in 2017 when Liu et al. published the global market projection on the healthcare workforce for 2030 [2]. In 2017, the World Health Organization (WHO) established a global strategy for human resources in health named Workforce 2030 [3]. As reported by Liu et al. [2], low- and middle-income countries face a lack of resources in delivering essential health services, and in 2020, the SARS-CoV2 pandemic exacerbated these needs. One possible solution is the development of assistive technologies that can help healthcare systems cope with these crises [2,4,5,6]. This is also important given the fact that according to the WHO, the number of older adults > 60 years old will increase to up to 1.4 billion by 2030 and up to 2.1 billion by 2050. The number of people with dementia is predicted to reach 75 million in 2030 [7]. In this picture, according to the global action plan of response to dementia, the development of assistive technologies like social assistive robots could be a strategic asset in managing the diagnosis, treatment, care, and support of people with dementia [7,8]. In 2022, Sorrentino et al. [9] highlighted how robotic technology can be integrated into individual care, enhancing the effectiveness and efficiency of healthcare services [9]. Pepper, a humanoid social robot developed by the Japanese Aldebaran (United Robotics Group) [10] company, is one of the most popular social robots available on the market. Introduced in 2014, Pepper is designed to interact with people in a natural and engaging manner, making it suitable for a variety of applications, including customer service, education, healthcare, and entertainment. Pepper is equipped with a set of sensors such as LED lights that change color to express different emotions, as well as cameras, microphones, and speakers, enhancing its communication capabilities. It has a touch-sensitive screen on its chest, allowing users to interact through touch gestures. In retail environments, the ability of the Pepper robot to recognize emotions is a critical aspect of its effectiveness in providing care. In 2022, D’Onofrio et al. [11] presented the “EMOTIVE Project”, which focused on emotion recognition by a Pepper robot, indicating a significant advancement in our knowledge of the robot’s empathetic capabilities so that it can better interact with patients. In recent years, social robots have been employed in different innovative research fields to enhance people’s well-being, autonomy, and independence [12]. Several studies have contributed to the understanding and validation of the feasibility and usability of social robots in various healthcare settings. In 2021, Cobo Hurtado et al. [13] developed and validated a social robot platform for physical and cognitive stimulation in elderly care facilities, demonstrating the benefits of such technology in enhancing care services. A study conducted by Asl et al. [12] improved the evidence-based methodology for using the MINI social robot with individuals with dementia and mild cognitive impairment. The study highlighted its potential impact on cognitive assessments and the provision of psycho-social and cognitive stimulation. In a previous study, the interaction with the robot was measured at the end of the intervention, which lasted for 1 month, with the Almere Model Questionnaire (AMQ) [14]. Usability and acceptability are essential factors when designing technology for users with mild cognitive impairment [15]. In 2020, Castilla et al. [16] conducted a usability study to evaluate the design of information and communication technology (ICT) for individuals with cognitive impairments, emphasizing the importance of user-centric approaches. Similarly, in 2018, Holthe et al. [17] conducted a systematic literature review to explore the usability and acceptability of technology for community-dwelling older adults with mild cognitive impairment and dementia, providing insights into the tailoring of technology to suit their needs. As the potential benefits of social robot interventions in mental healthcare continue to be explored, in 2022, Guemghar et al. [18] conducted a scoping review on the potential of social robot interventions in mental healthcare and identified the outcomes, barriers, and facilitators associated with their implementation. Moreover, some studies have focused on specific applications of social robots in cognitive impairment testing. In 2020, Martín Rico et al. [19] conducted an acceptance test for assistive robots, contributing to the understanding of how patients perceive and interact with robotic technologies. In 2020, Schüssler et al. [20] designed a study to evaluate the effects of a humanoid socially assistive robot compared to tablet training on the psycho-social and physical outcomes of persons with dementia, providing valuable insights into the potential benefits of robotic interventions.

### Objectives and Research Questions

The Mini-Mental State Examination (MMSE) [21,22] is widely utilized to screen for dementia and detect mild cognitive impairment. Our study aims to confirm whether it is possible for the Pepper robot to administer the MMSE and examine the results that are achieved. Following recent approaches [23] that integrate different technologies in robotic scenarios, this study is set up in the framework of the SocIal ROBOTics for active and healthy aging (SI-Robotics) project [24] and aims to test a robot equipped with an innovative technology that can acquire the psycho-physiological and emotional state of the patient during the execution of the test.

An approach based on the affective computing research paradigm can be fruitfully applied to this scenario. A social robot with the ability to recognize emotions or psycho-physiological states could provide the clinician with very detailed and important information. Moreover, giving the robot the capability of discerning the Arousal State (ArS) of the interlocutor could result in a better and more fluid interaction. Many scientific works in the last decade have investigated the use of affective computing in several fields of robotics from educational [25] to rehabilitative [26], as well as general social robotics [27]. The assessment of the ArS of the subjects can rely on several methods: speech recognition and analysis, physiological signal analysis, facial expressions, body posture, and gesture analysis [28,29,30].

Regarding the use of social robots and affective computing in the research field of aging, the most recent work, published in 2023 by Yoshii et al., focused on the early detection of mild cognitive impairment (MCI) through a conversation between the Pepper robot and the patient [8]. The conversation was not a specific examination, as it focused on prosodic and acoustic features, the duration of the response time, and jitter. Based on this, the authors were able to classify people as having no cognitive impairment or MCI [8]. In light of these studies, our research aims to contribute to the growing body of knowledge on the feasibility and efficacy of social robotic technology enriched with systems to detect psycho-physiological and emotional states. By building upon the existing evidence, we seek to evaluate the potential of social robots as a valuable tool for cognitive impairment testing and patient care, ultimately benefiting elderly individuals and the healthcare system as a whole.

The primary objective of this study is to evaluate the differences between the scores obtained in the MMSE test performed in the traditional way by a psychologist or other health professional and those obtained by administering the same test using the robotic system described in the following sections. As a secondary objective, we investigate the relevance of psycho-physiological and emotional aspects in the performance of the test. We also test the usability and user experience, as perceived by the patients involved in this study.

## 2. Materials and Methods

### 2.1. Experimental Protocol

We enrolled 20 patients aged over 60 with an Activity Daily Living (ADL) [31] index ≥4 and with a traditionally computed MMSE [21,22] score >18. The exclusion criterion was the inability to sign the informed consent. The experimental scenario was set up within the healthcare facilities of the Casa Sollievo della Sofferenza Research Hospital in San Giovanni Rotondo, Italy. More specifically, the patients were recruited from the hospitalized older adults in the Rehabilitation Medicine Unit. The clinical protocol was approved by the local Ethical Committee on the 14th of July 2021 with the N 111/CE code number. The experimental scenario consisted of the following phases:At first, we verified that each participant agreed to take part in the study by signing the informed consent for interacting with the Pepper robot and for video recording. A psychologist explained the purpose of the study and introduced the Pepper robot. The psychologist ensured that the patient met the inclusion criteria. As per the approved clinical protocol, patients underwent a set of questionnaires aimed at assessing various dimensions that could arise from their interaction with the Pepper robot in the context of the robotic MMSE administration scenario. The tests included the Activity of Daily Living (ADL) [31], Instrumental Activity of Daily Living (IADL) scale [32], Mini-Mental State Examination (MMSE) [21,22], Exton–Smith Scale (ESS) [33], Mini Nutritional Assessment (MNA) [34], Short Portable Mental Status Questionnaire (SPMSQ) [35], and Cumulative Illness Rating Scale Comorbidity Index (CIRS-CI) [36]. These tests were used to compute the Multidimensional Prognostic Index (MPI) [37], whose values range from 0 to 1, with the following risk classification scale:1.0 to 0.33 low prognostic mortality risk at 1 year (MPI-1);2.0.34 to 0.66 moderate risk (MPI-2);3.0.67 to 1.00 severe risk (MPI-3).Then, on a different day from the one on which the administration of the tests took place and in accordance with the needs of the clinical ward, each patient was introduced to the robot. Each participant was led into the room designated for the experiment. Patients with motor difficulties were assisted in reaching the setting in a wheelchair. The psychologist made the participant comfortable. The dialogue then continued with the administration of the MMSE test by the robot.The interaction with the robot was evaluated by administering 5 different tests to each participant at the end of the session. The tests were:1.The Almere Model Questionnaire (AMQ) to assess acceptability [14];2.The System Usability Scale (SUS) questionnaire to assess usability [38];3.The Robot Acceptance Questionnaire (RAQ) [39,40,41];4.The Godspeed test to assess likability [42,43];5.The User Experience Questionnaire (UEQ) [44,45].

This set of tests enabled a multifaceted evaluation of the robotic-mediated MMSE sessions. The questionnaires are described in more detail in Table 1.

To enable the Pepper robot to interact with the users, it was equipped with the RoboMate [46] system developed by Behaviour Labs [47]. RoboMate is a software platform run by humanoid robots like Pepper and it is recognized as a medical device for rehabilitation. The RoboMate app, through its graphic user interface in Figure 1, can help with the following activities:Simplify the use of robots by clinicians, therapists, and educators;Realize an easy and intuitive platform for human–machine interactions;Handle e-learning content and “edutainment”;Manage the delivery of content to the user;Track and store results of the executed sessions and patient data;Generate reports and statistics on the results of the executed sessions.

RoboMate is tailored to determine the behavior of a humanoid robot and hosts a tablet device on its chest as a reinforcing and feedback component through which a person can respond and interact during the session. The RoboMate system includes a mobile app for tablets, which enables the therapist to:Remotely control the movements and voice of the robot;Trigger predefined animations, games, and questions;Record answers;Tele-present a session using a mic and camera (Telepresence), in Figure 2, on the left;Manage patient data (sociodemographic, clinical data, and test session information), in Figure 2, on the right.

**Figure 1 biomimetics-08-00475-f001:**
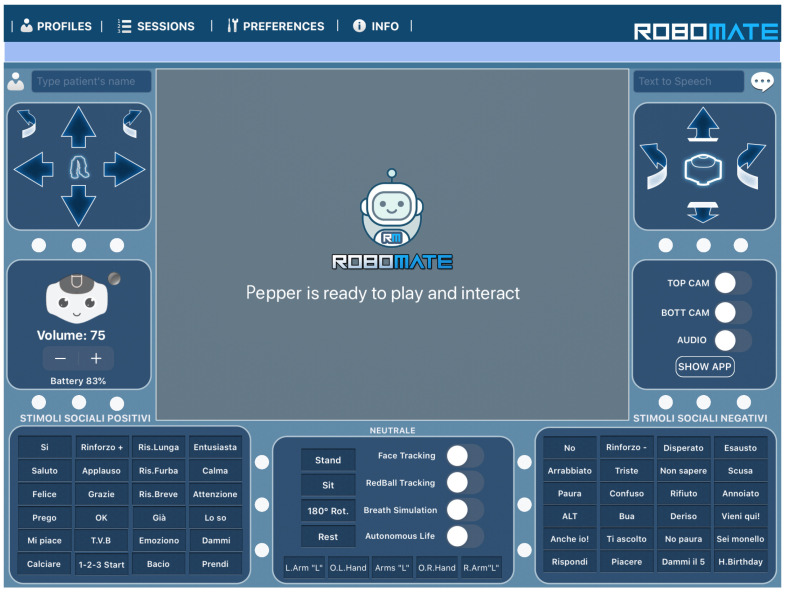
RoboMate app for remotely controlling the Pepper robot.

The psychologist then controls the robot with the tablet, which keeps the user engaged during the administration of the MMSE. Using its synthesized voice, the robot asks the user to answer the MMSE questions; as the user answers, the psychologist records the correctness of the answer on the tablet, as represented in the Figure 3.

The study included the evaluation of the patient’s psycho-physiological state during the MMSE performed by Pepper. For this purpose, a Computational Psychophysiology Module (CPM) was used, developed specifically for this purpose by Next2U. The CPM is a hardware platform consisting of an infrared (IR) image sensor, a visible (VIS) image recording device, and a computational unit based on the Jetson Nano system, which hosts artificial intelligence-based algorithms. The CPM was housed on the robot using a harness that allowed the vision module (IR + VIS sensors) to be positioned below the tablet that the Pepper robot housed on its trunk. From this position, the face of the patient, who was sitting about 1 m away from the robot, was framed. The control of the CPM was delegated to a remote interface controlled by the psychologist through a tablet, who was present in the room, together with the robot and the patient, to guarantee the success of the test. In particular, the interface allowed for the start of thermal IR video recording and framing control, as well as providing service information on the status of the CPM. The technical features of the acquisition module of the CPM are summarized in Table 2.

The CPM allowed for the synchronous acquisition of IR and VIS videos of the faces of the patients during the administration of the cognitive tests. The system has been validated and used in several research studies [27,48,49].

The cameras were rigidly mounted in a case, which fixed their relative positions.

The camera system was calibrated using stereoscopic calibration, which allowed for the transformation of the coordinates of the 2D VIS image space into the coordinates of the 2D IR image space. Due to the optical co-registration of the optics and a face alignment model, the CPM was able to extract 68 facial landmarks from the VIS image feed and project the set of points onto the IR domain [50].

The regions of interest (ROIs) of the subject’s face in the IR were detected as polygonal masks with a selection of landmarks as vertices. The ROIs used in this study were the region of the glabella and the region of the nose tip.

The preprocessing pipeline is illustrated in Figure 4.

In Figure 5, the Graphical User Interface (GUI) that controls the CPM is shown.

Importantly, 12 out of 20 individuals were considered for further analysis due to technical issues related to the acquisition or synchronization of the data.

The CPM module was equipped with algorithms from affective computing based on artificial intelligence and computer vision methods [51]. Specifically, for the purpose of the present study, the focus was on the estimation of valence and arousal of the Affective State (AS) of the subjects during the execution of the cognitive tests that relied on the circumplex model approach.

The circumplex model is a psychological framework used to understand and categorize human emotions, interpersonal relationships, and personality traits. It was developed by Russel [52] and has been widely used in the fields of psychology, counseling, and interpersonal communication.

The circumplex model represents the following concepts on a circular diagram, with two main axes:1.**X-Axis (Horizontal):** This axis represents the degree of activation or intensity of an emotion or trait. Emotions and traits can vary from low activation (calm, relaxed) to high activation (excited, anxious).2.**Y-Axis (Vertical):** This axis represents the valence or emotional tone of the emotion or trait. Emotions and traits can vary from positive valence (pleasant, happy) to negative valence (unpleasant, sad).

The circular diagram is divided into different sectors or quadrants, each representing specific emotions, traits, or interpersonal styles. The exact arrangement and labels of these sectors can vary depending on the specific model or theory being used, but they generally follow the principles of the circumplex model.

The algorithm for the estimation of the AS was built upon the implementations of the classifiers for the Autonomic Neural System (ANS) valence state and the ArS based on IR imaging.

A 1D time-series thermal signal, measured in counts, was retrieved from each ROI by averaging over the pixels within the ROI. The two thermal signals were then fed to the valence and ArS classifiers.

The valence classifier was based on a Support Vector Machine (SVM) with a linear kernel relying on the IR signal from the nose tip, which was highly sensitive to ANS activity. The classifier operated on overlapping windows of 20 s, with a delay of 2 s between adjacent windows. The classifier provided the estimated valence value of each block, with possible values being “sympathetic”, “parasympathetic”, or “NA” when no estimate could be made. Hence, each valence output state refers to the 20 s prior to the estimation. Notably, in this study, the “sympathetic” class was considered indicative of negative valence, whereas the "parasympathetic" class was associated with positive valence [53,54].

The classifier was trained on the following set of features of the 20 s window:Signal mean from the 1st third of the window;Signal mean from the last third of the window;Difference between the mean of the 1st and 2nd thirds of the window;Signal entropy;Ratio of the 95th percentile to the 5th percentile;First-order polynomial fit coefficients of the fit curve over the 2nd and 3rd thirds of the window;Second-order polynomial fit coefficients of the fit curve over the window;Ratio between the spectral power of the signal in the bands 0.04 Hz–0.15 Hz and 0.15 Hz–0.4 Hz.

The ArS algorithm was based on previous results reported by Kosonogov et al. [55]. The model feeds on the thermal signal coming from the nose tip and the glabella. The classifier operates on overlapping windows of 8 s, with a delay of 1 s between each window. The algorithm estimates the ArS by classifying the average 1st-time derivative of the difference between the thermal signal of the nose tip and that of the glabella over a period of 8 s.

The length of the window was selected to take into account the temporal delay of the thermal response associated with the increase in arousal conditions [25,56]. The derivative was computed from the slope of the minimum squares line fit of the z-normalized difference signal over a window of 8 s. The thresholds of the slope were defined using a data-driven approach to classify the arousal response (high, medium, and low ArSs).

The AS algorithm combines the valence state and the arousal state in a valence–arousal plane analog to the circumplex circle of effect [57]. The algorithm discriminates 6 states:High arousal—positive valence (excited state);High arousal—negative valence (tense state);Medium arousal—positive valence (focused state);Medium arousal—negative valence (cautious state);Low arousal—positive valence (calm state);Low arousal—negative valence (bored state).

To assign the AS, the algorithm takes the time at which the current ArS value is emitted and pairs it with the simultaneous valence value, providing an output update every 2 s, thus allowing for real-time AS monitoring.

The real-time AS classification pipeline is shown in Figure 6.

### 2.2. Descriptive Data Analysis for Usability Test Score

Statistical analysis was performed with R [58] version 4.2.23. The normal distribution of the population was assessed using the Shapiro–Wilk test. The results confirmed that due to the very small sample size, most of the parameters were not normally distributed, with some exceptions. Non-parametric tests are considered the best choice due to the sample size, despite some exceptions. The Mann–Whitney test was performed to assess any differences in distribution between the two groups, whereas the Kruskal–Wallis test was used in cases involving three or more groups. The Wilcoxon paired test was employed to assess significant differences in the MMSE scores when administered with and without the robot. Bivariate correlation analysis was conducted using the Spearman method due to the sample size. Some of the graphics and figures were created using Microsoft Excel (Microsoft Office Professional Plus 2016), whereas others were generated using the R software version 4.2.23.

### 2.3. Data Analysis Processing for the CPM

To ensure that the analysis remained unaffected by the particular question posed, due to the small sample size, the average effective response across all the questions was considered for each participant. The subsequent 10 s window from the question was divided into 5 non-overlapping 2 s segments to determine the average modulation of the signals and states following a question and detect the response pattern of the ANS following the interaction with the robot.

Notably, all the points for which the classification resulted in null values were discarded from the statistical analysis.

For each segment, the difference between the number of sympathetic and parasympathetic states was calculated; thus, the mean difference was calculated for all patients in the time segment under consideration. To determine the statistical significance of this data, the normalized mean of the number of sympathetic states and parasympathetic states per time segment was calculated and a Student’s *t*-test was performed.

Regarding the AS, the global effect of the human–robot interaction (HRI) experience was analyzed by averaging the affective response during a temporal window of 10 s following the question. Specifically, ASs characterized by medium or high arousal, particularly those showing positive valence, were considered significant states of an interaction characterized by attention on the part of the patient.

## 3. Results

We recruited 23 patients but we only included 20 patients in the analysis due to missing data. The total number of male users was 85 % (with a male/female rate of 17/3). The mean age was 75.35 ± 7.86 years. On average, the educational level in years was 9.95 ± 4.63. The educational level did not differ among the groups analyzed. The years of education did not seem to affect usability and acceptability, except for a moderate negative correlation in some domains in the AMQ: Perceived Enjoyment (PENJ) (ρ = −0.446) and Perceived Sociability (PS) (ρ = −0.528); the novelty domain in the User Experience Questionnaire (UEQ) (ρ = −0.462); and the Perceived Intelligence (PI) domain of Godspeed (ρ = −0.527), all with *p*-values < 0.05.

In Table 3, we summarize the demographic characteristics of the cohort, presenting functional and cognitive information. To validate the case study, the MMSE score obtained by the psychologist through the traditional method was compared with that acquired by the Pepper robot using the non-parametric Wilcoxon test for paired data. Obtaining a *p*-value = 0.111 is not significant compared to a Type I error α of 0.05. Therefore, no substantial differences were reported between the traditional administration of the MMSE test and its administration through the Pepper robot following the adopted protocol. This result confirms similar findings in the literature [59] and represents a first step toward validating a potential robotic system that can autonomously administer this type of test in the future, even without the direct involvement of a healthcare operator.

On average, the recruited patients had a low 1-year prognostic mortality risk. Table 3 presents the values obtained for some statistical variables (mean ± standard deviation (SD) or median (interquartile range (IQR))) concerning the responses provided by the patients to the administered questionnaires.

### 3.1. Almere Model Questionnaire

Regarding the AMQ, it can be concluded that the patients achieved an average score of around 3.00 for almost all constructs, showing higher average scores in the domain of Perceived Sociability (PS) and a notably very high value for Anxiety. It is important to note that in this construct, the scoring for Anxiety was reversed, meaning that higher values were associated with lower anxiety levels. There was an average score for the remaining domains, which fell within the range of 3.00 and 3.50, except for Perceived Adaptability (PAD), Perceived Enjoyment (PENJ), and Perceived Ease of Use (PEOU), but Intention to Use (ITU), Facilitating Condition (FC), and Social Presence (SP) were slightly lower in this case Table 4.

The Almere questionnaire employs a Likert scale, and the majority of the constructs are positively oriented, except for Anxiety and two other items, which needed reverse scoring [14]. The reliability measured by the Cronbach’s α test was above 0.7, except for Anxiety (α = 0.618) and Facilitating Condition (α = 0.398). In the case of Anxiety, the low score in terms of reliability was due to some users fearing they may have broken something without proper assistance. None of them perceived the robot as frightening. The same sensation was reported in the FC domain, where users often reported feeling unequipped to use it alone without proper professional support. The user perception can be explained in terms of the length of the interaction with the robot, as it lasted only as long as the test was administered. As a result, the patients did not perceive themselves as capable of autonomously using it.

In Figure 7, we can observe the interrelationship among the domains of the AMQ. The correlation matrix was calculated using the Spearman method due to the sample size. Positive correlations suggest that as one variable increases, the other tends to increase, while for negative correlations, as one variable increases, the other tends to decrease.

### 3.2. Godspeed

The Godspeed score measures the level of safety perceived by the patients, i.e., the perceived level of danger and comfort during the interaction with the robot [42]. This level of safety is expressed through opposing adjectives to which the patient responds, based on their perception of the robot, with a Likert scale score from 1 to 7. Patients perceived the Pepper robot as intelligent and likable, but they did not attribute anthropomorphic characteristics to it, considering it, in any case, an artificial entity Table 5. These results align with the Social Presence domain of the AMQ, as the items presented in that domain generally explore how users perceive the robot to be a human or real person.

### 3.3. Robot Acceptance Questionnaire

R The Robot Acceptance Questionnaire (RAQ), a test developed by the H2020 project Empathic (see [60]), is a test used to comprehensively measure both the overall and specific acceptability of a robot [39,40,41]. The test consists of a total of four clusters divided into six sections: Section 3 [40] includes four sub-sections: pragmatic quality (PQ), hedonic and robot identity quality (HQI), hedonic and feeling quality (HQF), and attractiveness (ATT). Clusters 3 and 4 encompass the other four sections, with a particular focus on Section 6, which investigates the robot’s speech and communication (Figure 11). These sections are evaluated on a Likert scale, ranging from 1 (strongly agree) to 5 (strongly disagree) [40]. The sections listed above include both positive and negative responses, and the scores for the negative items have been reverse-coded. This means that in the final scoring, low scores indicate positive evaluations of the robot, whereas high scores indicate negative evaluations. As shown in Table 6, the average scores were low for all the analyzed domains, which means that the robot was perceived as useful, effective, practical, clear, and controllable (pragmatic qualities); moderately original, creative, presentable, and aesthetically pleasing (hedonic qualities); and it evoked positive emotions and was capable of engaging the user.

In addition to the results presented earlier, the following figures (Figure 8, Figure 9, Figure 10, Figure 11, Figure 12, Figure 13 and Figure 14) show some graphs obtained in response to specific questions from the RAQ [39,40]. As mentioned earlier, the RAQ is divided into four main clusters [39]. Cluster 1 has the goal of collecting some socio-demographic information: Section 1 (ease of use and frequency of use of devices) and Section 2 (willingness to interact with the robot). The remaining clusters (3 and 4) are composed of Section 4 (aim to know the perceived age of the robot), Section 5 (items that evaluate which tasks participants would entrust to the robot or occupations: from 1(Not suitable at all) to 5 (Very suitable)), and the previously mentioned Section 6. This information can help us better understand the characteristics of the sample of patients involved in the study and their perspectives regarding the use of the robot.

Commenting on the results presented above, it appears that:The use of digital devices by patients is infrequent (Figure 8, left);Smartphones are the most commonly used digital tool. In general, patients reported that they did not know if the use of other digital devices was difficult or not (Figure 8, right).

There was a high number of patients who expressed a positive attitude toward the use of the robot and a willingness to interact with it (Figure 9);The robot’s occupations appear to be an interesting aspect: housework and welfare were the most quoted occupations for Pepper (Figure 10). The target population appears to have had difficulty in identifying a unique occupation for the robot.

The robot’s speech abilities appear to be a critical aspect (Figure 11);Although the willingness to interact with the robot was not influenced by the age attributed to it (*p*-value = 0.655), Pepper conveyed the idea of an anthropomorphic robot with a “youthful” appearance: four of the participants perceived it as a child and four of them perceived it as being aged between 10 and 20 years (Figure 14). There were also no significant differences between the perceived age and the impact on the willingness to interact.

**Figure 11 biomimetics-08-00475-f011:**
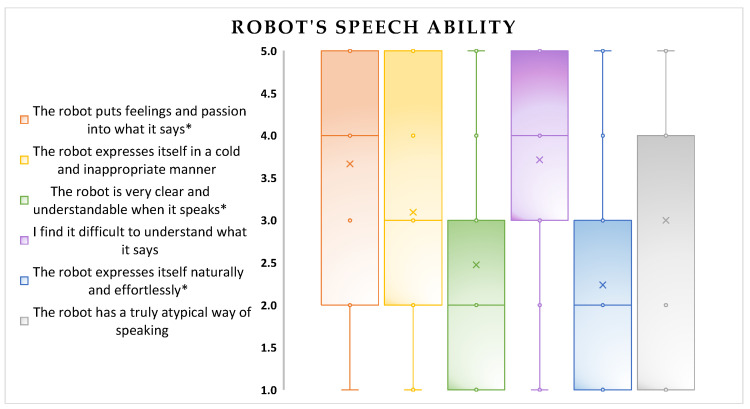
Scores of patient responses to RAQ Section 6 items. * Positive items reverse coded.

The above results cannot be extended to the entire elderly population because of the small number of patients involved in this study. However, they can be useful for characterizing the group of patients involved.
Figure 12Influence of robot’s age.
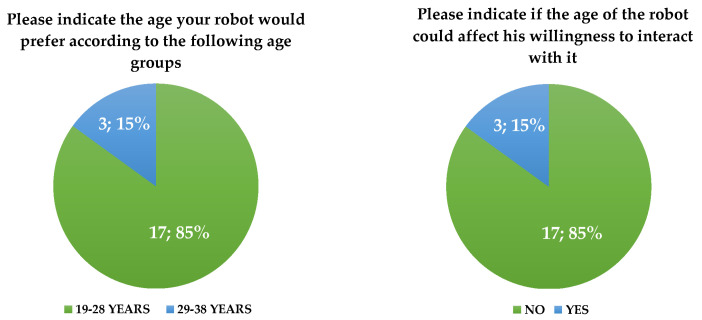

Figure 13Cross-tab between the two previous questions.
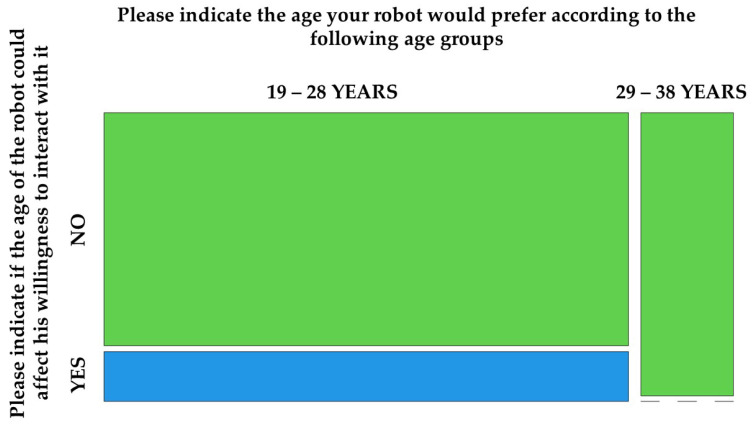

Figure 14Responses to the question: How many years do you attribute to the robot?
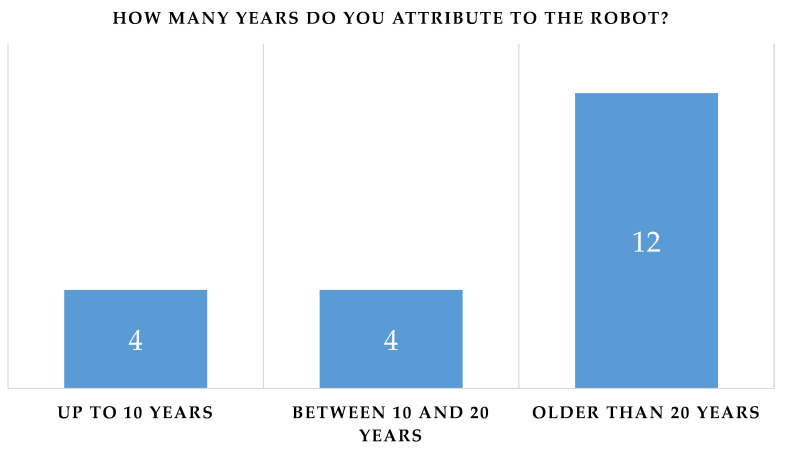


### 3.4. User Experience Questionnaire

The User Experience Questionnaire score does not produce an overall score for the user experience [44]. The scale ranges from −3 to +3, as reported in Table 7. In general, a value > +0.8 represents a positive evaluation [44]. In this case, for each domain, we obtained a positive evaluation, except for the stimulation (Table 7). In terms of reliability, we obtained a Cronbach’s α of below 0.7 for the domains of efficiency, dependability, and novelty. Interestingly, a moderate correlation between these domains and the SUS score is evident in Table 7, but it was weaker in the case of stimulation, attractiveness, and perspicuity. The participants did not perceive the robot as a novelty, but this seems not to have had a relationship with usability.

In Figure 15, the ranges of the User Experience Questionnaire (UEQ) scores are shown [44].

### 3.5. Differences in Usability among Patients’ Categories

Furthermore, an analysis was conducted using the non-parametric Kruskal–Wallis test to measure any differences in the distribution of the usability and acceptability test results based on the following categories:Patients’ levels of experience with technology;Patients’ genders;Cognitive status of the patients.

While no significance was found for the first two categories, regarding cognitive status, the following categories were considered:1.Cognitive impairment (CI; MMSE < 24.0);2.Mild cognitive impairment (MILD CI; MMSE < 27.0 and ≥ 24.0);3.No cognitive impairment (NO CI; MMSE ≥ 27.0).

No differences in the results on usability and acceptability were found to be significant for the cognitive categories. These results are reported in Table A1, Table A2, Table A3, Table A4, Table A5 and Table A6 in Appendix A. The data reported in Table A1 demonstrate how the patients were distributed in terms of cognitive status in relation to age, years of education, and gender. There were no differences between cognitive status and each category. The only clear difference that emerged among the three groups of patients with different levels of cognitive status was regarding the ESS test (*p*-value = 0.021) (Table A2). There were no differences in the scores of AMQ’s domains among groups with different cognitive statuses, as reported in Table A3. There were no differences among groups for the SUS score (*p*-value = 0.756).

The values recorded for the Godspeed test and the domains investigated by the Robot Assistant Questionnaire (RAQ) do not appear to be significantly different with respect to the cognitive status of the patients (Table A4 and Table A5). In the case of the User Experience Questionnaire, the score was the same across the three categories (Table A6).

The patients’ cognitive statuses do not seem to be a distinguishing factor in their interaction with the Pepper robot.

### 3.6. Differences in the Willingness to Interact with the Robot

We investigated the patients’ willingness to interact with the robot through the questions in Section 2 of the RAQ, in correlation with the UEQ, Godspeed, and domains outlined in Section 3 of the RAQ. Regarding the response categories for the willingness to interact, the groups were organized by consolidating the ‘Possible’ and ‘Probable’ responses into one category, and the ‘Improbable’ and ‘Impossible’ responses into another one. The scores ranged from 1 (Possible) to 5 (Impossible); we re-numbered them from 1 (Probable) to 3 (Improbable) for the subsequent analyses.

There was a statistically significant difference (*p*-value = 0.03) in the distribution between the genders of the participants and the willingness to interact with the robot, although the result could not be generalized due to the disproportionate number of males and females, as well as the sample size. No differences were highlighted between the willingness to interact among the cognitive test scores and in terms of years of education (*p*-value = 0.4832). The same result was found across the scores of the various domains of the AMQ. However, a negative correlation was found between certain domains of the AMQ and the willingness to interact with the robot (Table 8).

Table A7 shows the differences in the SUS scores for the willingness to interact. It is clear that the score is higher when the patient assumes that their interaction with the robot Pepper could be possible or probable rather than not possible or improbable. This trend is confirmed by a negative linear correlation between the SUS score and the willingness to interact (ρ = −0.670; *p*-value = 0.001).

The agreement between the willingness to interact with the robot and attractiveness, which represents the robot’s charm and appeal to the patient; efficiency, which is the perceived efficiency of the robot by the patient; dependability, which is the perceived reliability of the robot; and stimulation, which is how motivated the patient feels to use the robot, is statistically significant (Table A8). The higher the score, the greater the likelihood of interaction with the robot.

In Table A9, the results regarding the Godspeed domains are presented. In all the domains of Godspeed, the higher the score, the greater the likelihood of interaction with the robot. In Table A10, the results regarding the RAQ domains are presented. On the contrary, due to the inverse Likert scale, the lower the score, the greater the likelihood of interaction with the robot.

### 3.7. CPM Results

With regard to the valence, the *t*-test showed a significant difference between the sympathetic and parasympathetic valence during the 4–6 s segment following the questions (*p*-value = 0.001). For this time segment, the difference between the number of sympathetic and parasympathetic responses was 10.7% in favor of parasympathetic responses. For the rest of the temporal segments, no significant *p*-values emerged. The average differences are summarized in Figure 16.

Regarding the AS detection, the percentages of the occurrence of the estimation states are reported in Table 9.

It emerged that 94% of the ASs measured were indicative of medium or high arousal, 50.48% of the ASs measured were characterized by both medium-high arousal and positive valence, and 75.5% of the total number of states were characterized by medium arousal.

## 4. Discussion

This study investigated the feasibility of employing an assistive robot to administer cognitive tests in clinical practice. In particular, the robot Pepper was used to administer the MMSE to a geriatric population. The quality of the human–robot interaction (HRI) was monitored by administering questionnaires and evaluating the patients’ physiological responses. Specifically, the CPM was able to provide real-time monitoring of the valence, arousal condition, and AS.

The results demonstrated that Pepper was able to successfully administer the MMSE to patients since the scores obtained in the test were not statistically different from those obtained when the MMSE was administered through standard in-person delivery by a healthcare specialist. This result is in accordance with previous findings reported in the literature [59], highlighting the potentiality of the employment of assistive robots for the administration of cognitive tests. We found that there was no difference in the scores of acceptability and usability in the presence or absence of cognitive deficits. This could confirm that the cognitive status of the patients may not affect the usability and acceptability of the user experience.

The questionnaires administered showed the good acceptability of the artificial agent by the patients. In fact, the AMQ highlighted low levels of anxiety during the HRI, and the RAQ revealed the good acceptability of the robot. These results were confirmed by the CPM-based AS evaluation. This can be attributed to the fact that ASs such as anxiety can modulate the physiological state [61]: the prevalence of parasympathetic system activity with respect to sympathetic activity in the 4–6 s segment after the HRI is compatible with the hypothesis that between 4 and 6 s, after having received the question and elaborated on the answer, the patient feels comfortable with the administration of the test by the robotic agent. Moreover, 94% of the ASs measured are characterized by medium or high arousal, and 50.48% of the total affective conditions measured are denoted by both medium-high arousal and positive valence. Importantly, 75.5% of the total states are characterized by medium arousal, independently of the valence state. This can be related to the patient’s attention when listening to the questions asked by the robot and producing the answers. These estimated ASs showed a collaborative attitude aimed at carrying out the task requested by the robot.

Nevertheless, the willingness to interact with the robot had an important impact on the user experience. At the same time, the willingness to interact appeared to influence domains such as the perception of anthropomorphism, animacy, likability, perceived intelligence, and safety by the patient. It had an impact on the evaluation of the hedonic and pragmatic qualities of the robot and at the multi-level measure of the attractiveness that results from Pepper.

As shown in Table A9, patients with a probable willingness to interact appreciated the robot Pepper with respect to the others, attributing human-like qualities to it, and this benefited the overall interaction with the robot. Furthermore, contrasting resulted in the scores on likability, which was the highest among all levels of willingness to interact (Table A9), and it is not possible to determine whether the willingness to interact with the robot could be a cause or an effect of the perception of the robot’s intelligence.

However, it is worth noting that the Godspeed test demonstrated that although Pepper was perceived as likable and intelligent, it was not considered anthropomorphic by the patients. This aspect could be related to the low comprehensibility of the robot’s speech during the interaction, as shown by the RAQ Section 6 score concerning the robot’s speech abilities.

In general, among the population under examination, there seems to be unanimous agreement regarding the characteristics of the robot Pepper and the likelihood of future interaction with it. Perhaps in a larger sample, these differences would be accentuated or reduced. On the other hand, it appears that the willingness of a patient to interact with a new technology like a humanoid robot is a crucial factor for them to use the robot or perceive the experience as positive during the interaction. However, we can assume that the manifested anthropomorphism of the Pepper robot may be a key point in patients’ perceptions of the robot.

In a previous work by Szczepanowski et al. [62], the authors addressed the topic of perception toward social robots and their relationship with education. In our study and in our specific population, education did not influence the usability and the willingness to interact with the robot, but a lower educational level seemed to influence the perception of the novelty, intelligence, and perceived enjoyment and sociability of the robot. Other studies may confirm these findings.

### 4.1. Limitations

Two of the main limitations of this study are the poor intelligibility of the robot’s voice and the latency with which the robot sometimes reacted to the patient’s responses. These problems often led patients to ask the psychologist to repeat the question because they did not understand or were looking for feedback after having given the answer. This was reflected in the sporadic failure of the AS estimation due to signal losses by the CPM. Further studies should focus on the improvement of the spontaneity of the HRI and the fluidity of the communication to make the interaction more human-like.

Another limitation is related to the small study sample (low statistical power) and its gender imbalance. Moreover, the environmental setting in which the study was conducted may have influenced the performance of the patients, the HRI, and, consequently, the quality of the test administration. Hence, several environments should be used for testing to investigate the generalization of the results. From this perspective, it is worth highlighting that assistive robots could be employed in home environments, facilitating telemedicine and remote health monitoring. It is very important to investigate the replicability of the approach adopted in this study in a domestic setting. However, in the current state of development, specialized personnel are needed to properly control and manage the robot and the CPM. Therefore, further efforts should be directed to make the system user-friendly and suitable for non-trained users and caregivers for at-home usage.

Eventually, equipping the robot with the ability to perceive the AS of the patients and modify its behavior accordingly could enable the robot to offer emotional support to the patient and make the administration of cognitive tests more human-like.

### 4.2. Costs and Effectiveness

An analysis or evaluation that takes into account both the costs and effectiveness of the Pepper robot in a healthcare context could be of interest. The aim would be to determine whether resource allocation is efficient and whether the benefits obtained justify the costs incurred.

In the healthcare field, this type of analysis could be used to assess the relationship between the costs of the robot and the outcomes achieved in terms of the improved department and/or outpatient activity, as well as patient health, if applicable.

The goal would be to find an optimal balance between the costs incurred and the outcomes achieved to maximize the efficiency and effectiveness of using a robot like Pepper to deliver cognitive tests. In this context, an interesting experiment was conducted by D’Onofrio et al. [63] involving a humanoid robot that autonomously performed and managed the execution of the multidimensional assessment phase of the Comprehensive Geriatric Assessment (CGA), with the aim of assisting the healthcare professional [63,64].

Our results appear to be promising in a hospital and rehabilitation context, where the goal is to streamline and reduce diagnosis and follow-up times.

### 4.3. Future Perspectives

The results obtained in our study offer many paths of investigation to further develop the system and generate value from it. First, as digital skills are constantly progressing in our society, we expect future patients to obtain higher scores in all test domains, which can be a facilitating factor for the diffusion of this technology.

Our results open up the possibility of saving time for healthcare professionals. In fact, a robot that performs cognitive tests at different times, even in unusual settings, i.e., at home or far from clinics, will generate valuable data that clinicians can exploit when physically facing the patient in follow-up visits.

From a more research-oriented perspective, the availability of a CPM module coupled with a robot can pave the way for new studies on (a) the variability of MMSE results depending on environmental factors, and (b) tracking the emotions of a patient while a cognitive test is being executed.

Grasping patient emotion during an MMSE can also be useful for adapting the robot interaction by modifying its voice tone, speech, and intensity of movements in response to the emotional state of the patient so that a more natural and easy interaction can occur.

Lastly, the use of a social robot to administer tests like the MMSE could pave the way for the standardization of results so that they cannot be affected by the “dependence on the operator” characteristic, which is one of the issues with the traditional way of delivering cognitive tests.

## 5. Conclusions

This study demonstrated the feasibility of employing the Pepper robot equipped with the CPM for administering the MMSE in clinical practice. The results demonstrated the acceptance of the robotic system by the patients, supporting the hypothesis that robotic agents can be successfully used in such contexts. Moreover, the cognitive state did not affect the usage of the robot: the robot was generally appreciated for its likability and presumed age. Lastly, the different degrees of willingness to interact with the robot among patients align with their perceived acceptability, usability, and user experience.

Further studies should improve the spontaneity of the interaction, allowing the robot to adapt its actions autonomously in accordance with the AS of the patients. The findings of this study could pave the way for the large-scale employment of robots in both outpatient environments and for at-home usage.

## Figures and Tables

**Figure 2 biomimetics-08-00475-f002:**
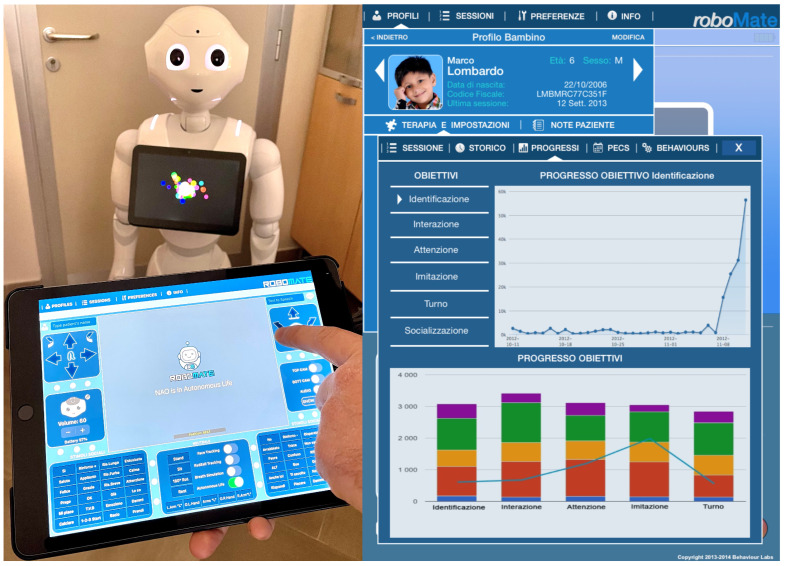
Managing users’ sessions remotely and viewing results on RoboMate: a screenshot.

**Figure 3 biomimetics-08-00475-f003:**
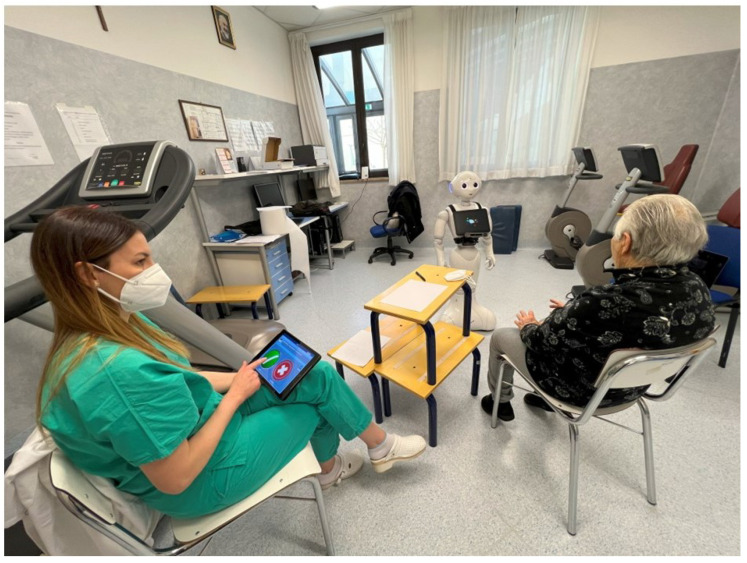
Mini-Mental State Examination using Pepper robot with the supervision of a psychologist.

**Figure 4 biomimetics-08-00475-f004:**
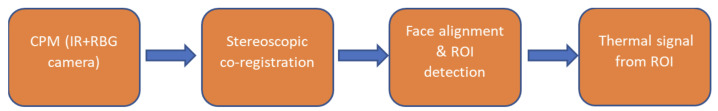
Preprocessing pipeline.

**Figure 5 biomimetics-08-00475-f005:**
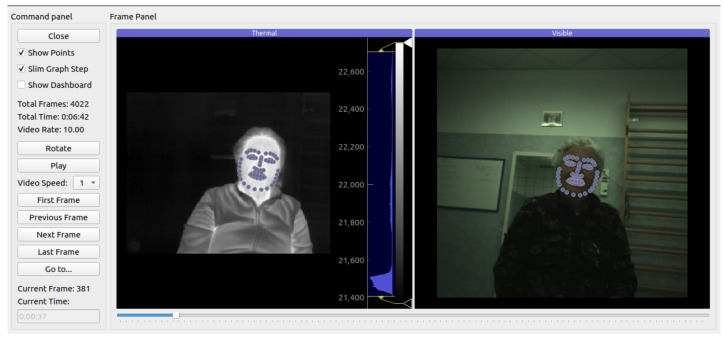
GUI of the CPM system. IR and VIS images with the fiducial 68 face landmarks are shown, respectively, in the left and right frames of the GUI.

**Figure 6 biomimetics-08-00475-f006:**
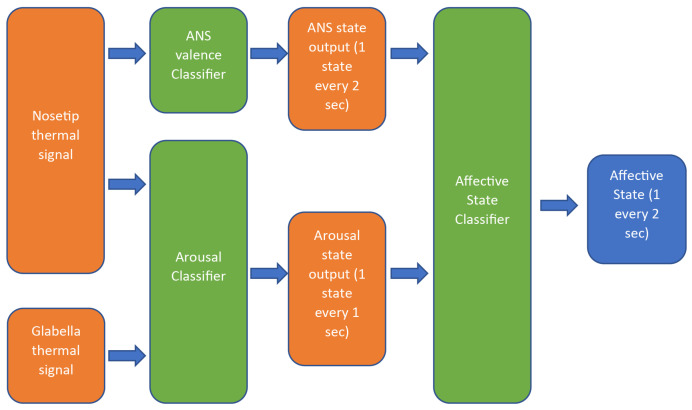
The realtime AS classification pipeline based on the valence and arousal classifiers, fed with thermal IR signals.

**Figure 7 biomimetics-08-00475-f007:**
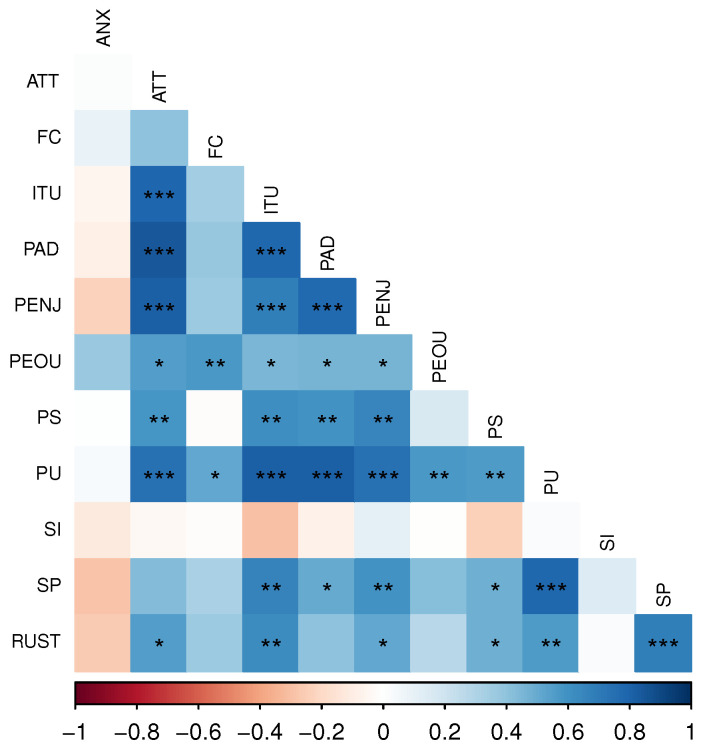
The correlation matrix among the domains of the AMQ: colors represent the degree of association between variables. Blue has been used to indicate positive correlations close to +1, while red is associated with negative correlations close to −1. * *p*-value < 0.05; ** *p*-value < 0.01; *** *p*-value < 0.001.

**Figure 8 biomimetics-08-00475-f008:**
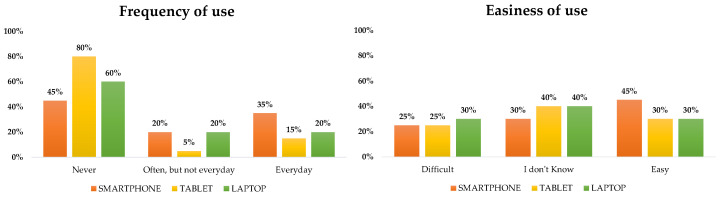
Familiarity of patients with digital devices with respect to the ease of use.

**Figure 9 biomimetics-08-00475-f009:**
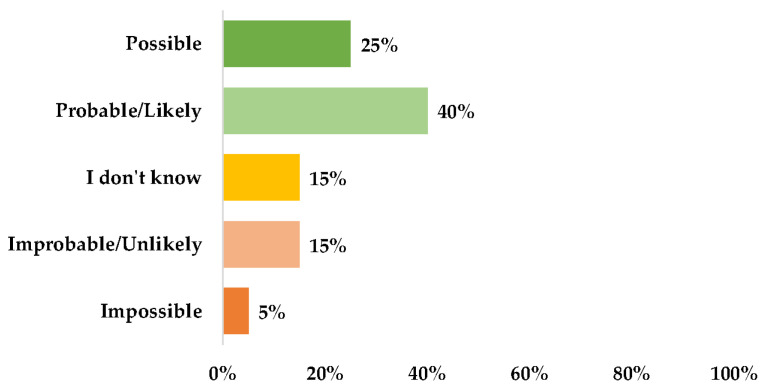
Willingness of the user to interact with the robot.

**Figure 10 biomimetics-08-00475-f010:**
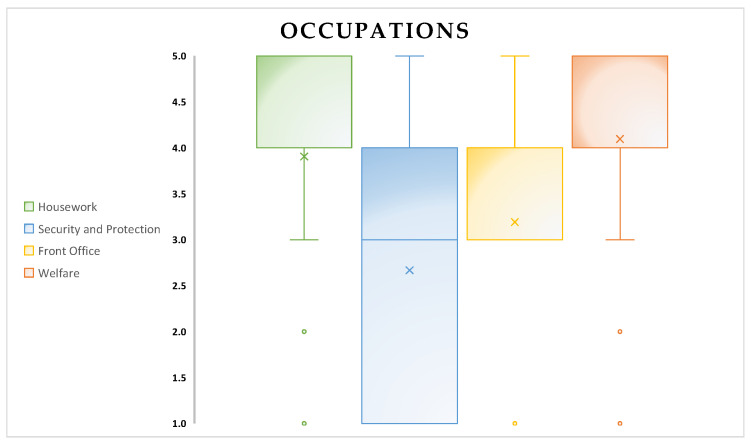
Occupations of participants that they would entrust a robot with.

**Figure 15 biomimetics-08-00475-f015:**
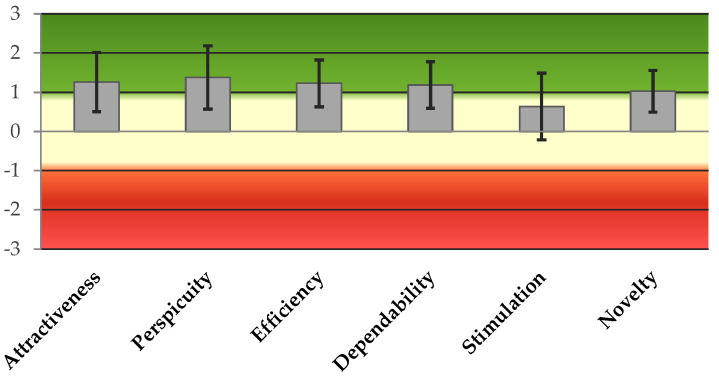
The ranges of scores for UEQ responses. The transformed score ranges from −3 (indicating extremely poor, in red) to +3 (representing exceptionally good, in green).

**Figure 16 biomimetics-08-00475-f016:**
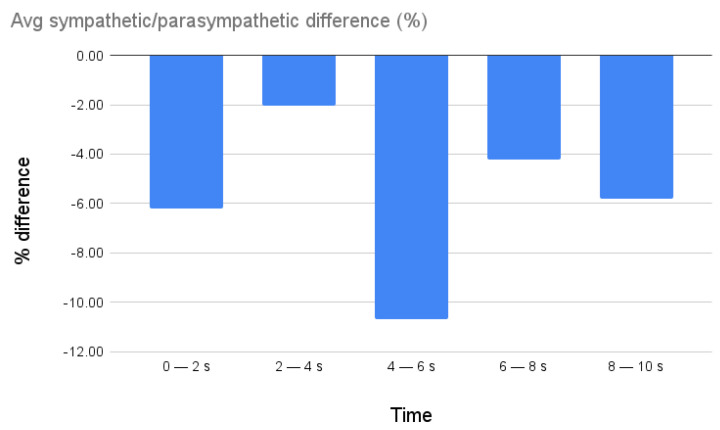
The average differences between sympathetic and parasympathetic responses.

**Table 1 biomimetics-08-00475-t001:** Standard questionnaires used to evaluate the interaction with the robot.

Questionnaire	Description
**Almere Model Questionnaire (AMQ)**	The questionnaire assesses the intention of use, anxiety, trust, enjoyment, and ease of use. The responses are measured on a Likert scale, with values ranging from 1 to 5, and then an average value for each domain is calculated [14].
**Godspeed**	The questionnaire evaluates the appearance and design of the robot in terms of anthropomorphism, animacy, likability, perceived intelligence, and perceived safety. The responses consist of opposing adjectives as items, per domain, and are measured on a Likert scale from 1 to 7 [43].
**Robot Acceptance Questionnaire (RAQ)**	This questionnaire evaluates the acceptance of the robot based on its pragmatic, hedonic, and attractiveness qualities, the attributed and perceived age of the robot, and the tasks it can perform. It is divided into different sections (at least 6) and scored on a Likert scale from 1 to 5 [39].
**User Experience Questionnaire (UEQ)**	The questionnaire aims to assess the pragmatic and hedonic quality of a specific product. Similar to Godspeed, it is designed with opposing adjectives as items, per domain, and is measured on a Likert scale from 1 to 7 [44].
**System Usability Scale (SUS)**	The SUS test is a ten-item questionnaire. The scores vary from 0 to 100 and are measured on a Likert scale (from 1 to 5). The SUS questionnaire is capable of acquiring a subjective assessment of usability. A value above 68 is considered acceptable [38].

**Table 2 biomimetics-08-00475-t002:** Technical features of the CPM sensing unit mounted on Pepper.

	VIS Device	IR Device
**Technical Data**	Intel RealSense D415	FLIR Boson 320 LWIR
**Weight**	4.54 g	7.5 g w/o lens
**Dimensions**	99 × 20 × 23 mm	21 × 21 × 11 mm w/o lens
**Spatial Resolution**	720 × 720 px	320 × 256 px
**Framerate**	10 Hz	10 Hz

**Table 3 biomimetics-08-00475-t003:** Demographic and cognitive characteristics of the cohort of 20 patients involved in the experiment.

Variables (Min–Max)	N. 20
**Traditional MMSE** (0–30)	26.35 [23.56–28.78]
**Robotic MMSE** (0–30)	26.20 [23.00–27.15]
**ADL** (4–6)	6 [6–6]
**IADL** (0–8)	8 [7.25–8.00]
**SPMSQ** (0–10)	1.00 [2.00–1.00]
**CIRS-CI** (0–3)	2.00 [1.75–3.00]
**MNA** (<17–≥24)	22.01 ± 2.13
**ESS** (5–20)	18.00 [18.00–18.00]
**MPI** (0–1)	0.17 [0.17–0.25]

**Legend:** Activity of Daily Living (ADL); Instrumental Activity of Daily Living (IADL); Mini-Mental State Examination (MMSE); Exton-Smith Scale (ESS); Mini Nutritional Assessment (MNA); Short Portable Mental Status Questionnaire (SPMSQ); Cumulative Illness Rating Scale Comorbidity Index (CIRS-CI); Multidimensional Prognostic Index (MPI). If data are normally distributed, mean ± SD is reported; otherwise, median (IQR).

**Table 4 biomimetics-08-00475-t004:** Results obtained through the Almere Model Questionnaire.

AMQ Items (Min–Max)	N. 20
**Anxiety** (ANX)	4.63 [3.94–5.00]
**Attitude** (ATT)	3.22 ± 1.25
**Facilitating Conditions** (FC)	2.30 ± 0.91
**Intention to Use** (ITU)	2.83 [1.00–4.00]
**Perceived Adaptability** (PAD)	3.50 [1.58–4.17]
**Perceived Enjoyment** (PENJ)	3.40 ± 1.27
**Perceived Ease of Use** (PEOU)	3.13 ± 1.01
**Perceived Sociability** (PS)	3.71 ± 1.08
**Perceived Utility** (PU)	3.03 ± 1.13
**Social Influence** (SI)	3.00 [1.38–4.00]
**Social Presence** (SP)	1.80 [1.00–2.80]
**Trust**	3.50 [1.00–4.00]

If data are normally distributed, mean ± SD is reported; otherwise, median (IQR).

**Table 5 biomimetics-08-00475-t005:** Results of Godspeed test.

Godspeed’s Domains	N. 20
**Anthropomorphism (ANTP)**	2.10 [1.55–3.25]
**Animation (ANM)**	2.80 ± 1.11
**Likeability (LIKE)**	4.60 [3.40–4.85]
**Perceived Intelligence (PI)**	4.00 [3.40–5.00]
**Perceived Safety (PSa)**	3.66 [2.83–3.66]

If data are normally distributed, mean ± SD is reported; otherwise, median (IQR).

**Table 6 biomimetics-08-00475-t006:** Results obtained from the RAQ (Robot Acceptance Questionnaire).

RAQ’s Domains	N. 20
**Pragmatic Quality (PQ)**	2.35 [1.75–3.38]
**Hedonic Quality—Identity (HQ-I)**	2.50 ± 0.94
**Hedonic Quality—Feeling (HQ-F)**	2.15 [1.35–2.90]
**Attractiveness (ATTr)**	2.59 ± 0.97

If data are normally distributed, mean ± SD is reported; otherwise, median (IQR).

**Table 7 biomimetics-08-00475-t007:** Results obtained for the UEQ (User Experience Questionnaire) and correlation with the SUS.

UEQ’s Domains	N. 20	Cronbach’s α	SUS’s Spearman ρ	*p*-Value
**Attractiveness**	1.92 [0.13–2.58]	0.853	0.490	*
**Perspicuity**	2.12 [0.75–2.75]	0.829	0.540	*
**Efficiency**	1.23 ± 1.38	0.542	0.700	***
**Dependability**	1.50 [0.63–2.00]	0.423	0.720	***
**Stimulation**	0.64 ± 1.95	0.827	0.600	**
**Novelty**	1.03 ± 1.21	0.159	0.290	

**Legend:** System Usability Scale (SUS). If data are normally distributed, mean ± SD is reported; otherwise, median (IQR). The transformed score ranges from −3 (indicating extremely poor) to +3 (representing exceptionally good). * *p*-value < 0.05; ** *p*-value < 0.01; *** *p*-value < 0.001.

**Table 8 biomimetics-08-00475-t008:** Spearman correlation between the domains of the AMQ and the willingness to interact with the robot.

Domains of the AMQ	Spearman ρ	*p*-Value
**Attitude (ATT)**	−0.580	**0.008 ****
**Intention to use (ITU)**	−0.560	**0.011 ***
**Perceived Adaptability (PAD)**	−0.570	**0.009 ****
**Perceived Enjoyment (PENJ)**	−0.590	**0.006 ****
**Perceived Utility (PU)**	−0.500	**0.026 ***

**Legend:** * if *p*-value < 0.05; ** if *p*-value < 0.01.

**Table 9 biomimetics-08-00475-t009:** Occurrence of the estimation states.

Negative Valence	Positive Valence
High Arousal and Negative Valence (Tense) = 7.92%	High Arousal and Positive Valence (Excited) = 9.57%
Medium Arousal and Negative Valence (Cautious) = 35.60%	Medium Arousal and Positive Valence (Focused) = 40.91%
Low Arousal and Negative Valence (Bored) = 6.01%	Low Arousal and Positive Valence (Calm) = 0.00%

## Data Availability

The data presented in this study are available on request from the corresponding author. The data are not publicly available due to restrictions (they contain information that could compromise the privacy of research participants). Samples of the compounds are available from the authors.

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
