# Peer review of "Assessing Feasibility of Cognitive Impairment Testing Using Social Robotic Technology Augmented with Affective Computing and Emotional State Detection Systems"

_biomimetics, 2023, doi:10.3390/biomimetics8060475_

Round 1

Reviewer 1 Report

The article is well written, well organized and easy to read, presenting an interesting research on the usage of a robot to evaluate a person according to a cognitive scale.

The main problem found was the confusion with AS as Affective State and Arousal State. In many places AS is meaning Affective State, but it is defined two times is Arousal State.

Another problem with emotional evaluation is that, dominance is not the same as valence. Emotional evaluation can be defined in terms of valence, arousal and dominance. Valence ranges from happy to unhappy. Arousal ranges from calm to excited. Dominance ranges from controlling the situation to not controlling the situation.

It is mandatory to correct the use of affective state, valence, arousal and dominance in the text.

My concerns are about the concepts, not the English language.

Author Response

The article is well written, well organized and easy to read, presenting an interesting research on the usage of a robot to evaluate a person according to a cognitive scale.

We want to thank the Reviewer for the effort put in revising our manuscript. Below are reported the responses to the Reviewer’s comments and the manuscript was revised accordingly. We do believe that following his advice and comments, the manuscript gained more scientific value.

The main problem found was the confusion with AS as Affective State and Arousal State. In many places AS is meaning Affective State, but it is defined two times is Arousal State.

We thank the Reviewer for pointing out this mistake. We now considered AS for Affective State and ArS as Arousal state. The manuscript has now been modified accordingly.

Another problem with emotional evaluation is that, dominance is not the same as valence. Emotional evaluation can be defined in terms of valence, arousal and dominance. Valence ranges from happy to unhappy. Arousal ranges from calm to excited. Dominance ranges from controlling the situation to not controlling the situation.

We thank the Reviewer for highlighting this aspect. We totally agree with this observation and, accordingly, we modified dominance with valence in the manuscript, since valence is more appropriate in the context.

It is mandatory to correct the use of affective state, valence, arousal and dominance in the text.

We thank the Reviewer for this comment. We modified the manuscript using AS for Affective State and ArS for Arousal State.

Reviewer 2 Report

Dear Authors,

Thanks for giving me a chance to read this manuscript, “Assessing Feasibility of Cognitive Impairment Testing Using Social Robotic Technology Augmented with Affective Computing and Emotional State Detection Systems”. The current paper validate the Mini-Mental State Examination (MMSE) test administered by the Pepper robot equipped with systems to detect psychophysical and emotional states.

This is an interesting and solid work in the field of digital health. However, there are minor issues in the current manuscript that should be addressed.

1.      Literature and method

·        Since trust is a significant dimension, authors are advised to argue the measurement used in this study and the latest research (Song et al., 2023).

Ref:

Song, Y., Tao, D., & Luximon, Y. (2023). In robot we trust? The effect of emotional expressions and contextual cues on anthropomorphic trustworthiness. Applied ergonomics, 109, 103967.

To sum up, I personally like this paper. However, the problems should be addressed in order to be further considered. Hope these suggestions help.

Sufficient enough

Author Response

Dear Authors,

Thanks for giving me a chance to read this manuscript, “Assessing Feasibility of Cognitive Impairment Testing Using Social Robotic Technology Augmented with Affective Computing and Emotional State Detection Systems”. The current paper validate the Mini-Mental State Examination (MMSE) test administered by the Pepper robot equipped with systems to detect psychophysical and emotional states. This is an interesting and solid work in the field of digital health. However, there are minor issues in the current manuscript that should be addressed.

 We want to thank the Reviewer for the effort put in revising our manuscript. Below are reported the responses to the Reviewer’s comments and the manuscript was revised accordingly. We do believe that following his advice and comments, the manuscript gained more scientific value.

  1. 1.      Literature and method

  • Since trust is a significant dimension, authors are advised to argue the measurement used in this study and the latest research (Song et al., 2023).

Ref:

Song, Y., Tao, D., & Luximon, Y. (2023). In robot we trust? The effect of emotional expressions and contextual cues on anthropomorphic trustworthiness. Applied ergonomics, 109, 103967.

To sum up, I personally like this paper. However, the problems should be addressed in order to be further considered. Hope these suggestions help.

We thank the Reviewer for this comment. We added the proposed reference in the manuscript.